# Uridine Treatment of the First Known Case of SLC25A36 Deficiency

**DOI:** 10.3390/ijms22189929

**Published:** 2021-09-14

**Authors:** Luisa Jasper, Pasquale Scarcia, Stephan Rust, Janine Reunert, Ferdinando Palmieri, Thorsten Marquardt

**Affiliations:** 1Department of Pediatrics, University Hospital of Münster, Albert-Schweitzer-Campus 1, Gebäude A13, 48149 Münster, Germany; luisa.jasper@uni-muenster.de (L.J.); stephan.rust@ukmuenster.de (S.R.); janine.reunert@ukmuenster.de (J.R.); 2Department of Biosciences, Biotechnologies and Biopharmaceutics, University of Bari, Via Orabona 4, 70125 Bari, Italy; pasquale.scarcia@uniba.it

**Keywords:** SLC25A36, SLC25A33, pyrimidine nucleotide carrier, mitochondrial solute carrier, uridine, bypass therapy

## Abstract

SLC25A36 is a pyrimidine nucleotide carrier playing an important role in maintaining mitochondrial biogenesis. Deficiencies in SLC25A36 in mouse embryonic stem cells have been associated with mtDNA depletion as well as mitochondrial dysfunction. In human beings, diseases triggered by *SLC25A36* mutations have not been described yet. We report the first known case of SLC25A36 deficiency in a 12-year-old patient with hypothyroidism, hyperinsulinism, hyperammonemia, chronical obstipation, short stature, along with language and general developmental delay. Whole exome analysis identified the homozygous mutation c.803dupT, p.Ser269llefs*35 in the *SLC25A36* gene. Functional analysis of mutant SLC25A36 protein in proteoliposomes showed a virtually abolished transport activity. Immunoblotting results suggest that the mutant SLC25A36 protein in the patient undergoes fast degradation. Supplementation with oral uridine led to an improvement of thyroid function and obstipation, increase of growth and developmental progress. Our findings suggest an important role of SLC25A36 in hormonal regulations and oral uridine as a safe and effective treatment.

## 1. Introduction

The solute carrier superfamily (SLC) consists of more than 400 different membrane proteins that transport ions as well as loaded and unloaded molecules. The SLC superfamily is divided into 65 families. The 53 members of the SLC25 family, also known as mitochondrial carrier family (MCF), work as uni- and anti-porters, transporting various solutes over the inner mitochondrial membrane [1,2], such as ATP, ADP, amino acids, cofactors, tricarboxylic acid cycle intermediates and pyrimidine nucleotides [1,3]. These solutes are necessary for energy production, RNA and DNA synthesis and mitochondrial function in general [4,5]. The members of the SLC25 family share a common structure [2,6,7]. Their primary structure is formed by three tandem repeats, each consisting of two transmembrane alpha helices, which are connected by large hydrophilic regions. The *N*- and *C*-termini are located on the cytosolic side of the membrane [8]. Solute carrier family 25 members 33 (SLC25A33) and 36 (SLC25A36) are the only known mitochondrial pyrimidine nucleotide carriers in humans [9]. They share a homologous sequence with 61% amino acid identity [10]. Especially, SLC25A36 plays an important role in maintaining mitochondrial biogenesis [11]. Human diseases related to SLC25A36 are not yet known. SLC25A36 depletions in mouse embryonic stem cells (mESC) have been associated with mtDNA depletion as well as mitochondrial dysfunction [11].

There are already two inherited disorders of pyrimidine biosynthesis known, developmental and epileptic encephalopathy-50 caused by CAD-mutations (MIM: 616457) and hereditary orotic aciduria (MIM: 258900), both leading to a lack of substrate to be transported into mitochondria. Uridine itself as a pyrimidine replacement is known as a safe and effective treatment of hereditary orotic aciduria [12]. Uridine as a therapy for CAD mutations has also been proven to be effective by Koch et al. [13]. In addition, oral application of deoxy pyrimidine monophosphates has been shown as a bypass therapy for thymidine kinase 2 deficiency, a disease which leads to unbalanced deoxynucleotide pools [14].

Here, we report on the first patient with SLC25A36 deficiency. The 12-year-old girl presented with hypothyroidism, hyperinsulinism, hyperammonemia, chronical obstipation, short stature, along with language and general developmental delay. Therapy with uridine led to normal thyroid function, loss of obstipation, normoglycemia, growth increase and less aggressive behavior.

## 2. Results

### 2.1. Patient and Clinical Phenotype

The girl is the second child of consanguineous Turkish parents. Pregnancy was uneventful. Due to a suspected stroke of the mother, the child was born in the 38th week by caesarean section. She was born with a weight of 2605 g (11th percentile), 45 cm length (1st percentile) and 33.5 cm head circumference (30th percentile). Her older brother, a 17-year-old boy, is healthy.

When the girl was five months old, hyperammonemia (93 µmol/L, reference 16–53 µmol/L) and hypoglycemia (2.5 mmol/L, reference > 3.33 mmol/L) were noticed for the first time while she was being treated in a hospital for pneumonia. When she was 11 months old, an injury led to another hospitalization. She presented with high levels of ammonia (160 µmol/L) as well as a hypoglycemia (1.89 mmol/L). This time, pathological levels of insulin were noted (52.5 pmol/L, reference < 21 pmol/L). Hyperinsulinism was diagnosed, and no mutations were found by genetic analysis. Treatment with diazoxide was started. Nonetheless, hypoglycemic-induced seizures led to hospitalization on five occasions until today. A temporary discontinuation of diazoxide at the age of three years showed high insulin levels up to 164.5 pmol/L in hypoglycemia.

Primary hypothyroidism was diagnosed at the age of 13 months, when she was first investigated for growth retardation. TSH was 8.79 mIU/L (reference 0.6–5.7) with low fT4, and treatment with levothyroxine was initiated. An ultrasound at the age of 2 years showed a hypoplastic left thyroid flap (0.07 mL) and a normal sized right thyroid flap (0.72 mL) (reference whole volume 1.5 ± 1.4 mL). Despite treatment, TSH levels remained elevated with levels between 6.63 and 13.48 mIU/L, on average 7.81 mIU/L (general reference 0.8–5.3 for girls < 12 years, also depending on age and laboratory). At all times, our patient was substituted with levothyroxine up to 6 µg/kg body weight. Controls were performed at least three times a year. Ammonia levels were also never normal, the values were measured between 80 and 189 µmol/L, on average 120.45 µmol/L (reference 16–53 µmol/L). Elevation of liver enzymes (GOT, GPT, gammaGT, alkaline phosphatase) was not observed at any time. Protein restriction as well as therapy with citrulline showed no decline of ammonia levels and citrulline was not tolerated by the patient.

At the age of four years, an expressive language retardation as well as a general developmental delay were diagnosed. IQ testing showed an IQ of 58 in SON-R and 55 in IDS. She attends a special needs school. Furthermore, ADHD was diagnosed two years later, which is treated with methylphenidate. Regarding her stunted growth (below the 10th percentile), normal levels of growth hormones were found, and a family background was associated (mother: 155 cm, father: 160 cm).

She was sent to our clinic at the age of eight years, where she presented with hypothyroidism, hyperinsulinism, hyperammonemia, hypercholesterolemia, chronical obstipation, stunted growth, expressive language retardation, general developmental delay and ADHD.

Exome sequencing identified eleven homozygous variants in the patient (available in the Supplementary material, Appendix A), which were deemed potentially causative based on the reported consanguinity. Following exclusion of unlikely candidates based on allele frequency or presence in our in-house database of variants, as outlined in the Methods Section, homozygous variants in the genes *SLC25A36* (MIM: 616149), *TUBA4A* (MIM: 191110) and *CRACR2B* (MIM: 614177) were identified as possible candidates. Further in silico prediction analysis [15] paired with correlational analysis considering the clinical phenotype rendered the homozygous truncating variant c.803dupT (p.S268Ilefs*35 (Ref-Seq ENSG00000324194; NM_001104647.3)) in *SLC25A36* the most likely candidate. Both parents were heterozygous for the variant. In addition, compound heterozygous variants in two genes were detected but eventually excluded based on allele frequency in publicly available repositories. Full sequencing data from the index patient as well as the parents are available upon reasonable request.

An analysis of her amino acid profile showed no abnormalities, and glutamine was always within the reference range. In the physical examination, a small atrial septal defect without hemodynamic relevance was found as well as a physiological ECG. Her skull MR showed a small volume loss of the brain, mainly in the frontal parts, while the pituitary gland presented normally. A whole-body MR showed age-appropriate results, especially of the pancreas. In an EEG, no indications for increased susceptibility to seizures were found—her seizures have probably been caused by hypoglycemia. An ultrasound of the thyroid gland showed a proper structure with a normal right (3.6 mL volume) and a clearly smaller left flap (0.4 mL) (reference whole volume 4.2 ± 1.4).

A muscle biopsy (M. vastus lateralis) showed a normal amount of mtDNA, with a mtDNA/nDNA ratio of 184% of the average of the reference group, as well as ATP and creatine phosphate production within the reference range. At the enzyme level, normal activities of the respiratory chain enzymes, complex V and citrate synthase were measured. For light and an electron microscopy, muscle was again used, and normal muscle tissue was apparent with only an isolated increase of glycogen. ATPase-staining with different pH levels showed no grouping of fibers, and in a COX/SDH-staining, no COX-negative fibers could be found, as well as no ragged red fibers in modified Gomori trichrome staining. Acid phosphatase activity was not increased.

### 2.2. Functional Analysis of the SLC25A36 p.Ser269llefs*35 Mutant

To assess the pathogenic potential of the *SLC25A36* c.803dupT mutation, the activities of wild-type (WT) and p.Ser269llefs*35 *SLC25A36* were assayed in liposomes reconstituted with each of these proteins, as uptake of ^3^H-CTP (the best substrate of SLC25A36 [3]) into proteoliposomes in exchange for intraliposomal unlabeled CTP. The WT SLC25A36 protein efficiently catalyzed ^3^H-CTP/CTP exchange, whereas it was negligible if any transport activity was observed with the p.Ser269llefs*35 mutant SLC25A36, even after a long period of incubation (Figure 1A,B), despite normal insertion of the mutant protein in the liposomal membrane. The percentages of ^3^H-CTP transported into p.Ser269llefs*35 proteoliposomes at the initial rate and after 60 min were about 1.1 and 1.6 respectively, as compared to those taken up by control proteoliposomes in 4 experiments, showing that the transport activity of the p.Ser269llefs*35 mutant SLC25A36 is virtually abolished.

In another set of experiments, the presence of p.Ser269llefs*35 SLC25A36 (with a calculated molecular mass of 33.36 kDa) and WT SLC25A36 (with a calculated molecular mass of 34.22 kDa) in the patient fibroblasts was tested with an anti-SLC25A36 polyclonal antibody. No SLC25A36 or its mutant form was detected in fibroblasts of the patient, whereas a single immunoreactive band with an apparent molecular mass of about 34 kDa was observed in the fibroblasts of two unrelated healthy controls (Figure 2). In contrast to SLC25A36, the β-subunit of the ATP synthase complex was present in approximately equal amounts in the fibroblasts of the patient and controls (Figure 2). These results suggest that the p.Ser269llefs*35 mutant undergoes fast degradation in the patient’s cells. Notably, the anti-SLC25A36 polyclonal antibody used in this study was able to detect the p.Ser269llefs*35 SLC25A36 mutant protein after its expression in E. coli substantiating the absence of this protein in the patient’s fibroblasts.

### 2.3. Response to Uridine Treatment

During uridine therapy, TSH levels rapidly declined with high levels of fT3 and fT4, indicating that her thyroid gland immediately started working on its own. Figure 3 shows the development of TSH under the phasing out process of levothyroxine treatment. Diazoxide was phased out without hypoglycemia. Furthermore, the child’s chronical obstipation stopped immediately under uridine treatment. The patient’s mother also reported that her daughter behaves less aggressively and is more patient. According to the mother, our patient also has less language restrictions since she suddenly started to speak words she did not seem to know before.

After two and a half months without levothyroxine treatment, TSH rose again to 8.53 mIU/L (reference 0.28–4.3). Supplementation of 200 µg of iodine each day resulted in TSH, fT3 and fT4 within the reference range. Six months after the first ultrasound of the thyroid gland, a new investigation showed a significant growth of both flaps. The right flap grew from 3.6 to 7.2 mL volume, and the left one from 0.4 to 0.7 mL (reference whole volume 4.4 ± 2.1). The growth was accompanied by a further improvement of TSH levels.

Due to the impaired glutathione pathway in mESC [11], administering 500 mg of vitamin C per day was started to prevent oxidative stress. It was tolerated by the patient without any side effects.

In the clinical follow-up, cholesterol was always within the reference range, but ammonia showed no continuous decline. An increase of growth was observed nine months into therapy (Figure 4). IQ testing one year after the start of the treatment showed an IQ of 49 on the Wechsler Intelligence scale for children, fifth edition. No side effects were noticed at any time.

## 3. Discussion

Di Noia et al. [3] showed that SLC25A36 transports (deoxy)nucleoside mono-, di- and tri-phosphates (dNMPs, dNDPs, dNTPs) of the bases cytosine (C), uracil (U), inosine (I) and guanin (G), with a preference for C and U, by uniport and antiport. SLC25A33 transports the bases thymine (T), C and U as (deoxy)nucleoside di- and triphosphates, with a preference for uridine and thymidine phosphates. Both mitochondrial carriers do not transport adenine nucleotides [3]. Physiologically, SLC25A36 has the major activity [3]. Consistent with our patient’s chronical obstipation and general developmental delay, Xin et al. [11] described a high expression of SLC25A36 in the large intestine and the brain. SLC25A36 is supposedly not expressed in the liver [11], but nevertheless, our patient presented with hyperammonemia. The expression in the thyroid gland and pancreas was not analyzed by Xin et al., and therefore further studies on the function of SLC25A36 in these organs are needed. According to the GTEx Portal (https://gtexportal.org, website accessed on 26th of August 2021), SLC25A36 is expressed in the thyroid gland. Furthermore, protein expression in the pancreas is described.

Pyruvate dehydrogenase complex defects and mtDNA depletion in muscle were excluded. This fits the patient since she is not having any musculoskeletal symptoms.

Consistent with the patient’s general developmental delay, SLC25A36 is highly expressed in the cerebral cortex and furthermore in the brainstem and pineal gland [10]. The GTEx Portal describes, amongst other tissues, an expression in the gastrointestinal tract, but not in the liver.

Furthermore, from this data, it becomes clear that except for the liver, SLC25A36 tends to be expressed at higher levels than SLC25A33.

The homozygous mutation c.803dupT, p.Ser269llefs*35 in *SLC25A36* was identified by whole exome analysis of the child and her parents. It deletes the sixth transmembrane domain of the protein, so a non-functioning protein could be expected. This was proven in functional analysis of the *SLC25A36* p.Ser269llefs*35 mutant in proteoliposomes. Besides its loss-of-function, the mutant protein undergoes quick degradation in the patient’s cells.

Xin et al. [11] analyzed SLC25A36 deficiencies in mESC. SLC25A36′s transport steps are necessary for synthesis and breakdown of RNA and mtDNA. It was shown that the relative amount of mtDNA was reduced in SLC25A36 knockdown mESC [11]. Variations in mitochondrial size, number and mass were observed [11]. SLC25A36′s function is necessary for mitochondrial genome maintenance and regulation of mitochondrial membrane potential. Thus, SLC25A36 deficiencies result in mitochondrial dysfunction as well as a reduced mitochondrial membrane potential [11]. The depletion of mtDNA still has to be shown in human cells.

Glutathione works as an antioxidant in its reduced form (GSH). It can be oxidized into a disulfide (GSSG). In SLC25A36 knockdown mESC, GSH levels were largely decreased, while GSSG levels were increased, resulting in a reduced GSH/GSSG ratio, indicating oxidative stress [11]. This potential oxidative stress in the patient was addressed by administering oral vitamin C. Its supplementation to uridine treatment should be extended and eventually combined with other antioxidative substances.

Uridine is an essential part of de novo pyrimidine synthesis as well as the salvage pathway. In several steps, uridine-monophosphate (UMP) is synthesized de novo from glutamine and bicarbonate via the CAD enzyme complex. Figure 1 describes the further modification of UMP to (deoxy-) thymidine-monophosphate (TMP) by thymidylate synthase (TS), as well as the transport steps of SLC25A33 and SLC25A36. UMP can also be modified into cytidine-monophosphate (CMP). In the salvage pathway, uridine needs to be deoxygenated via ribonucleotide reductase (RNR) and phosphorylated to dUMP. The first steps of de novo synthesis and the salvage pathway take place in the cytoplasm. The modification of dTMP to dTTP can take place in mitochondria (Figure 1). In the literature, it is disputed whether TS exists in mitochondria [16,17] or not [18]. Even RNR might also be located in mitochondria [19]. Hence, the modification of UMP might directly take place inside the mitochondria.

Even though our patient’s pyrimidine biosynthesis has no defects, restrictions in her mitochondrial DNA-synthesis in the organs with SLC25A36 expression can be expected, most probably resulting in a lowered amount of mtDNA [11]. That can be attributed to the fact that there are less pyrimidines transported into her mitochondria.

The second mitochondrial transporter of the SLC25 subfamily, SLC25A33, was addressed by uridine to stimulate pyrimidine uptake into mitochondria. Uridine was chosen since SLC25A33 prefers uridine phosphates as substrates, with SLC25A33 expression suppression leading to lowered mitochondrial UTP levels [20]. SLC25A33 expression can be induced by insulin-like growth factor 1 (IGF-1) and insulin when cells need energy to support an altered phenotype, stimulating mitochondrial biogenesis [20]. This suggests an upregulation of SLC25A33 to compensate the loss of SLC25A36′s function. Furthermore, uridine can penetrate the blood–brain barrier by equilibrative nucleoside transporters [21].

Overexpression of SLC25A33 leads to increased levels of mtDNA [22], indicating that addressing SLC25A33 can compensate for the lowered mtDNA due to SLC25A36 deficiency.

Treatment with oral uridine led to a fast response, while no side effects were noticed at any time. Prior to uridine treatment, the patient’s TSH was always above normal levels, except for three times, and euthyroidism could not be achieved even with levothyroxine doses up to 6 µg/kg body weight. TSH levels normalized fast when uridine was administered, indicating a normal thyroid function. This development was reinforced by providing oral iodine daily, resulting in a significant growth of the prior small thyroid gland. Furthermore, she was normoglycemic at all times while on uridine. From the first day of treatment, her chronical obstipation stopped. Her behavior at home and in school improved. Normal ammonia levels could not be achieved at any time by administering uridine, in line with the fact that SLC25A36 is not expressed in the liver [11]. Further research is necessary to clarify the cause of the hyperammonemia. A second independent problem was not found by whole exome analysis.

To see if the amount of mtDNA increases when uridine is administered would provide a better understanding of the exact mechanisms of the effect of uridine treatment.

The increase of growth nine months into therapy (Figure 4) could not be noticed to the same extent in the following months. Growth development has to be observed over time.

Further studies about the exact mechanisms of SLC25A36 deficiencies will advance the understanding of its physiological functions. More cases of SLC25A36 deficiencies will help us to see the effects in humans. Noticing this disease in patients as early as possible so a treatment with oral uridine can be started rapidly will give these patients a better outcome. Oral uridine can be seen as a safe and effective treatment of SLC25A36 deficiencies.

In conclusion, we have reported on the first patient described with SLC25A36 deficiency. The phenotype is characterized by hypothyroidism, hyperinsulinism, hyperammonemia, chronical obstipation, short stature, as well as language and general developmental delay. A therapy with oral uridine ameliorates the symptoms, leads to normal thyroid function and increases the quality of life of the patient.

## 4. Materials and Methods

### 4.1. Genetic Analysis and Clinical Diagnostics

Blood samples were obtained from the patient and her parents, and exome sequencing as trio analysis was performed as described previously [23,24]. Written informed consent for the genetic analysis was obtained prior to analysis. Data analysis was approved by the local ethics committee (Ethikkommission der Ärztekammer Westfalen-Lippe, No. 2019-199-f-S).

Several diagnostic steps were taken to analyze the effects of the SLC25A36 deficiency. Instrumental diagnostic ECG, EEG, ultrasound of the thyroid gland and skull and whole-body MR were used. A muscle biopsy (M. vastus lateralis) was performed to analyze the amount of mtDNA [25] as well as the function [26] of mitochondria in muscle.

### 4.2. Bacterial Expression, Reconstitution into Liposomes and Transport Assays of Recombinant WT SLC25A36 and p.Ser269llefs*35 SLC25A36 Mutant

To conduct functional studies of WT versus p.Ser269llefs*35 *SLC25A36* using a bacterial system, plasmids were constructed containing the coding sequence of WT *SLC25A36* amplified by PCR from human brain cDNA. The mutation c.803dupT was introduced using the QuikChange^®^ Site-Directed Mutagenesis Kit with CAGATCTTTTTTTCAGACTCTTATCTTTGCTTGTTCAAGAAG as a forward primer and CTTCTTGAACAAGCAAAGATAAGAGTCTGAAAAAAAGATCTG as a reverse primer. Constructs were transformed into *Escherichia coli* TOP10 cells. WT and p.Ser269llefs*35 SLC25A36 were overexpressed as inclusion bodies in the cytosol of *E. coli*, solubilized, purified and reconstituted into liposomes, as previously described [27,28,29]. Notably, equivalent amounts of each recombinant protein were used for their in vitro reconstitution, and the amount of both proteins incorporated into liposomes was about 18% of the protein added to the reconstitution mixture [3,30,31,32]. Then, the transport activities in proteoliposomes reconstituted with recombinant SLC25A36 or p.Ser269llefs*35 SLC25A36 mutant were measured as ^3^H-CTP/ATP exchange after 3 min of incubation (in the initial linear range of substrate uptake to measure the specific activity) and after 60 min of incubation, using the efficient and rapid transport stop-inhibitors, 10 mM of pyridoxal 5′-phosphate and 20 mM of bathophenanthroline, as previously described [33,34,35].

### 4.3. Immunoblotting

Western blots were performed using antibodies against SLC25A36 and the β-subunit of the F1-ATP synthase complex, purchased from Invitrogen, Waltham, MA, USA (cod. PA5-31606) and BD Biosciences, Franklin Lakes, NJ, USA (cod. 612519), respectively. Densitometric analyses were accomplished using the Image Lab™ Touch software (Bio-Rad Laboratories, Hercules, CA, USA).

### 4.4. Uridine as a Therapeutic Intervention

After informed consent, uridine was administered as four oral daily doses (100 mg/kg bodyweight per day), following the dosage of Koch et al. [13]. The treatment was started when the patient was 10 years and 3 months old. Oral uridine in pharmaceutical quality was imported from FORMULA LABOR Schaffhausen, Switzerland.

## Data Availability

Not applicable.

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
