# Peer review of "Uridine Treatment of the First Known Case of SLC25A36 Deficiency"

_ijms, 2021, doi:10.3390/ijms22189929_

Round 1
Reviewer 1 Report
The manuscript by L. Jaspert et al. reports the first known case of SLC25A36 deficiency. The clinical phenotype of this new disease has been accurately described over the years in the affected 12-years-old girl. The key symptoms of the disease are hypothyroidism, hyperinsulinism, hyperammonaemia, chronical obstipation, short stature along with language and general developmental delay. The molecular defect of the disease has been carefully disclosed. Direct transport measurements of the S269Ifs*35 mutant SLC25A36 and wild type SLC25A36 reconstituted into liposomes under the same conditions revealed that the transport activity of the S269Ifs*35 SLC25A36 is virtually abolished. Furthermore, evidence has been provided that the variant protein undergoes fast degradation in the patient’s cells. Although the authors did not attempt to investigate the etiopathology of the main symptoms, they have properly discussed the relationships between these symptoms and tissue-specific expression of SLC25A36 (mainly deduced from Human Protein Atlas) as well as the role of the SLC25A36-mediated transport abilities and properties (deduced from the literature and from studies in SLC25A36 knockdown mESC). Finally, the authors have described in details a safe and highly efficient treatment of the disease based on supplementation of uridine. The authors have in fact documented well that during uridine therapy almost all the main symptoms of the disease were relieved or dismissed. Again the relationships between the modifications of uridine (and its derivatives) in the cells and in the mitochondria and the transport steps catalyzed by SLC25A36 and SLC25A33 (i. e., the second mitochondrial transporter of pyrimidine nucleotides in humans) have been well explained. Although further research is needed on other aspects of SLC25A36 deficiency, the work of L. Jasper is a great leap forward our complete understanding of the mitochondrial carriers associated diseases. In summary this study is scientifically solid, well written, novel, interesting and significant. For these reasons I recommend the publication of this manuscript in IJMS.
Minor comment
Line 41 and some other places: the word “mitochondrial” is missing in relation to carrier(s) or transporter(s).
Author Response
Dear reviewer,
Thank you for your comment. I inserted the word „mitochondrial“ in relation to transporter and carrier in line 41 and other fitting places.
Kind regards,
Luisa Jasper
Reviewer 2 Report
Luisa Jasper and colleagues report on a single patient with a homozygous frame-shift variant in SLC25A36. The affected 12-year-old girl presented with developmental delay, intellectual disability, chronical obstipation, short stature, hypothyroidism, hyperinsulinism, and hyperammonemia. SLC25A36 protein was shown to be absent in patient fibroblasts and the recombinant truncated protein did not show transport activity. Most remarkably, treatment of the patient with oral uridine improved several clinical parameters within 22 months of supplementation.
This manuscript is of high interest, especially the obvious response to uridine treatment looks impressive and spreading of this finding by publication is definitely important. The major weakness of the study is the limited number of just a single patient, however, the clear workup of the underlying pathomechanism and the obviously effective treatment are convincing enough to support publication of this manuscript. Prior to a final decision there are, however, some points that should be improved:
- An illustration of the genetic finding is lacking. Please either provide results of NGS trio sequencing or targeted Sanger sequencing from the patient and parents.
- There is no reference sequence number (GenBank etc.) for the identified variant in SLC25A36 given. Please add.
- The mutation nomenclature, especially at the level of protein is chaotic and erroneous. Partially the wild-type amino acid is missing, on page 3 line 115 you use the term "SLC25A36 SC269Ifs*35"! Furthermore, I would strongly recommend to use the 3-letter code for amino acids, e.g. "p.Ser269Ilefs*35" or alternatively "p.Ser269IleTer35" throughout the manuscript.
- Do you think that elevated ammonia is due to the SLC25A36 deficiency or is it a second independent problem of this patient? Has there any elevation of liver enzymes been observed? Do you consider protein restriction?
- Throughout the manuscript reference ranges for normal values are insufficiently provided, especially in the case report section 2.1.
- Do you think that treatment with vitamin c has any positive effect? Do you plan to continue or extend and eventually combine with other antioxdative substances?
- The legend to the metabolic scheme is incomplete, at least concerning the first word. Furthermore, the asterisk should be explained.
- Throughout the manuscript, please insert a blank between numbers and units, e.g. "2605[blank]g (11th percentile)".
- When discussing the expression levels of SLC25A36 and SLC25A33, I would recommend to refer to GTEX (https://gtexportal.org/home/gene/SLC25A36 and https://gtexportal.org/home/gene/SLC25A33). From these data is comes clear that the expression level of SLC25A36 tends to be higher than that of SLC25A33 despite of liver (as has been cited from reference 11). Furthermore, using these data, I would recommend to skip the speculation on page 7 line 200 that SLC25A36 is not expressed in muscle.
- Page 3, lines 107-108, "creatine phosphate" should be typed in lower case.
Page 3, line 113, "trichrom[e]"
Author Response
Dear Reviewer,
Thank you very much for your comments. Please see the attachment.
Kind regards,
Luisa Jasper
